# Molecular and Clinical Features of Pancreatic Acinar Cell Carcinoma: A Single-Institution Case Series

**DOI:** 10.3390/cancers16193421

**Published:** 2024-10-09

**Authors:** Ashwathy Balachandran Pillai, Mahmoud Yousef, Abdelrahman Yousef, Kristin D. Alfaro-Munoz, Brandon G. Smaglo, Jason Willis, Robert A. Wolff, Shubham Pant, Mark W. Hurd, Anirban Maitra, Huamin Wang, Matthew Harold G. Katz, Laura R. Prakash, Ching-Wei D. Tzeng, Rebecca Snyder, Luca F. Castelnovo, Anthony Chen, Andrey Kravets, Kseniia Kudriavtseva, Artem Tarasov, Kirill Kryukov, Haoqiang Ying, John Paul Shen, Dan Zhao

**Affiliations:** 1Department of Hospital Medicine, The University of Texas MD Anderson Cancer Center, Houston, TX 77030, USA; abalachandran1@mdanderson.org; 2Department of Gastrointestinal Medical Oncology, The University of Texas MD Anderson Cancer Center, Houston, TX 77030, USAabyousef@salud.unm.edu (A.Y.); kdalfaro@mdanderson.org (K.D.A.-M.); bgsmaglo@mdanderson.org (B.G.S.); jason.willis@mdanderson.org (J.W.); rwolff@mdanderson.org (R.A.W.); spant@mdanderson.org (S.P.); mwhurd@mdanderson.org (M.W.H.); lfcastelnovo@mdanderson.org (L.F.C.); achen10@mdanderson.org (A.C.); jshen8@mdanderson.org (J.P.S.); 3Ahmed Center for Pancreatic Cancer Research, The University of Texas MD Anderson Cancer Center, Houston, TX 77030, USA; 4Department of Pathology, The University of Texas MD Anderson Cancer Center, Houston, TX 77030, USA; amaitra@mdanderson.org (A.M.); hmwang@mdanderson.org (H.W.); 5Department of Translational Molecular Pathology, The University of Texas MD Anderson Cancer Center, Houston, TX 77030, USA; 6Department of Surgical Oncology, The University of Texas MD Anderson Cancer Center, Houston, TX 77030, USA; mhgkatz@mdanderson.org (M.H.G.K.); lrprakash@mdanderson.org (L.R.P.); cdtzeng@mdanderson.org (C.-W.D.T.); rsnyder@mdanderson.org (R.S.); 7BostonGene Corporation, 100 Beaver St, Waltham, MA 02453, USA; andrey.kravets@bostongene.com (A.K.); ksenia.kudryavtseva@bostongene.com (K.K.); artem.tarasov@bostongene.com (A.T.); kirill.kryukov@bostongene.com (K.K.); 8Department of Molecular and Cellular Oncology, Division of Basic Science Research, The University of Texas MD Anderson Cancer Center, Houston, TX 77030, USA; hying@mdanderson.org

**Keywords:** *KRAS*, immunohistochemistry, acinar cell carcinoma, pancreatic, OS

## Abstract

**Simple Summary:**

We present a case series of 16 patients with pancreatic acinar cell carcinoma treated at our institution for whom available molecular information was evaluated. Most of the patients had metastatic disease, and all patients tested for *KRAS* mutations were *KRAS* wild type. Five of 12 patients who underwent DNA damage repair gene testing had germline and/or somatic mutations. One patient was found to have RET fusion and responded favorably to selpercatinib for over 42 months. We also include two additional cases who underwent BostonGene testing, including genomic alterations, RNA expression, and tumor microenvironment (TME) features. One of the two cases was found to have NTRK1 fusion. These findings highlight the need for further investigation of acinar cell carcinoma using larger samples to refine treatment strategies for this rare pancreatic cancer.

**Abstract:**

**Objectives:** Acinar cell carcinoma (ACC) accounts for about 1% of pancreatic cancers. The molecular and clinical features of ACC are less characterized than those of pancreatic ductal adenocarcinoma. **Methods**: We retrospectively evaluated the clinical and molecular features of ACC patients who underwent germline and/or somatic molecular testing at The University of Texas MD Anderson Cancer Center from 2008 to 2022 and two cases from 2023–2024 who underwent RNA and TME analysis by Boston Gene. Patient information was extracted from our institutional database with the approval of the Institutional Review Board. **Results:** We identified 16 patients with available molecular testing results. Fourteen patients had metastatic disease, one had borderline resectable disease, and one had localized resectable disease at diagnosis. Fifteen patients were wild type for *KRAS* (one patient had unknown *KRAS* status). Somatic/germline mutations of DNA damage repair genes (*BRCA1/2*, *PALB2*, and *ATM*) were present in 5 of 12 patients tested for these genes. One patient was found to have RET fusion and responded favorably to selpercatinib for over 42 months. The median overall survival (OS) was 24 months for patients with metastatic disease. One of the additional two cases who underwent BostonGene testing was found to have NTRK1 fusion. RNA and TME analysis by Boston Gene of the two cases reported immune desert features and relatively lower RNA levels of CEACAM5, CD47, CD74, and MMP1 and higher RNA levels of CDH6 compared with PDAC.

## 1. Introduction

Acinar cell carcinoma (ACC) is rare, accounting for about 1–2% of pancreatic cancers, and it has distinct clinical and molecular features [1]. Because of low ACC case numbers, treatment guidelines for this cancer are limited. Authors have reported better survival and greater chemosensitivity of ACC compared to pancreatic adenocarcinoma [2]. A retrospective analysis demonstrated longer survival in pancreatic cancer patients harboring actionable molecular alterations who underwent molecularly matched therapies in comparison with pancreatic cancer patients who did not have actionable molecular alterations [3]. Furthermore, the clinical and molecular features of ACC differ from those of pancreatic adenocarcinoma [4]. However, these features of ACC are not well understood. With the development of precision oncology and increased uptake of tumor molecular profiling, including next-generation sequencing (NGS), understanding the mutations associated with ACC is critical for the development of targeted therapy and novel treatment strategies.

In this study, we present a case series of 16 patients treated for ACC at The University of Texas MD Anderson Cancer Center (MD Anderson). We evaluated the molecular and clinical characteristics of these patients to improve our knowledge of the clinical and molecular features of ACC and suggest possible indications for treatment. We also included additional two cases who underwent Boston Gene testing, including genomic alterations, RNA expressions and tumor microenvironment (TME) features.

## 2. Materials and Methods

Sixteen patients with ACC who underwent molecular testing at MD Anderson from 2008 to 2022 were identified retrospectively. Tumor genomic alteration information, including *KRAS* mutation information, along with demographic, clinical, molecular, and pathological information was extracted from the institutional Palantir Foundry data science platform using natural language processing, with the approval of the MD Anderson Institutional Review Board (IRB). The information was validated via chart review, which included a review of each patient’s age at diagnosis, sex, presenting symptoms at diagnosis, tumor site, pathology and cytology results, epidemiological data, treatment details (surgery, radiation therapy, and/or chemotherapy), and status at last follow-up visit. Overall survival (OS) duration was calculated from the date of initial diagnosis to the date of death or last follow-up visit. The Kaplan–Meier method was used to estimate the median OS, and the Gehan–Breslow–Wilcoxon test was used to compare OS.

We also included two additional cases who underwent BostonGene testing, including genomic alterations, RNA expression, and tumor microenvironment (TME) features under another IRB in collaborative efforts with BostonGene. These two patients underwent testing in 2023 and 2024. The BostonGene Molecular Functional Portrait™ (MF Portrait™) integrates data obtained using the comprehensive genomic profile (CGP) and shows the composition and activity of both the malignant and the TME components. Gene signature profiling of >10,000 cancer patients identified four distinct tumor microenvironment subtypes conserved across 20 different solid cancers: immune-enriched, fibrotic (IE/F); immune-enriched, non-fibrotic (IE); fibrotic (F); and immune desert (D). The IE subtype is associated with a superior response to immune checkpoint inhibitors (ICIs) in cutaneous melanoma, bladder cancer, lung adenocarcinoma, and gastric cancer. The D subtype and IE/F subtype with low tumor mutational burden (TMB) are associated with poor response to ICIs in cutaneous melanoma and bladder cancer [5]. Visualization of tumor and microenvironment components in projection to a cohort of patients with a similar diagnosis, as well as the most significant alterations, was performed based on RNA expression profiling, which may differ slightly from DNA data, and RNA expression levels of selective genes were compared with PDAC reference cohort cases. 

## 3. Results

### 3.1. Demographic Characteristics

In this case series of 16 pancreatic ACC patients who underwent molecular profiling at MD Anderson, 14 patients had pure ACC, 1 patient had ACC mixed with neuroendocrine tumor, and 1 patient had ACC with ductal adenocarcinoma. The demographic features of these patients are shown in Table 1.

### 3.2. Clinical and Molecular Characteristics of ACC

The primary tumor was in the head of the pancreas in 50% of the patients (N = 8), the body of the pancreas in 31% of the patients (N = 5), and in the tail of the pancreas in 13% of the patients (N = 2). The tumor involved both the head and body of the pancreas in one patient. The most common symptom was abdominal pain or discomfort, which occurred in 12 patients; three patients presented with obstructive jaundice. At the time of initial diagnosis, 14 patients had metastatic disease, 1 had borderline resectable disease, and 1 had localized resectable disease (Table 1). Ten patients had liver metastasis at diagnosis. Five patients had regional lymph node and vascular involvement, making their tumors unresectable at the time of diagnosis.

The immunohistochemical staining profiles for the 16 study patients was summarized in Table 2. ACC demonstrated positivity for markers associated with acinar cell differentiation, such as trypsin (N = 9/9) and chymotrypsin (N = 2/2); epithelial origin, such as cytokeratin 7 (N = 8/9); and neuroendocrine differentiation, such as synaptophysin (N = 5/12), alpha 1-antichymotrypsin (N = 2/2), and chromogranin (N = 3/13). Additionally, expression of other markers, such as BCL10 (N = 2/2), BCL2 (N = 2/2), and alpha 1 antitrypsin (N = 1/2) was observed.

### 3.3. Genetic Alterations in ACC

Fifteen patients (94%) were wild type (WT) for *KRAS*, with one patient having unknown *KRAS* status. Somatic and/or germline mutations of DNA damage repair genes (*BRCA1/2*, *PALB2*, and *ATM*) were present in 5 of 12 patients (42%) tested for at least some of these genes. The most frequent genomic alteration was the mutation of *NF1*, followed by *CTNNB1* and *SMAD4*. Of note, one of the patients had *SAT-B1-RET* fusion. Due to the rare nature of pancreatic ACC and the extended period of study, the available genetic alterations of the patients are shown as an oncoplot in Figure 1.

### 3.4. Treatment

The details of treatment received by our patients are shown in Table 3. Six patients underwent first-line treatment with 5-fluorouracil, irinotecan, and oxaliplatin (FOLFIRINOX). Three of these patients had partial responses, two had stable disease, and one had progressive disease. Also, five patients received treatment with 5-fluorouracil and oxaliplatin (FOLFOX), with one patient having a response, two having stable disease, and two having progressive disease. Furthermore, five patients underwent treatment with gemcitabine-based regimens. Only one patient had a response after receiving gemcitabine/nab-paclitaxel combined with cisplatin. The other four patients who received gemcitabine-based regimens did not respond.

Fourteen patients had metastatic disease, one had borderline resectable disease, and one had localized resectable disease at initial diagnosis. The one patient who had resectable disease underwent the Whipple procedure without neoadjuvant or adjuvant chemotherapy and 2 years later presented with liver metastasis, which was treated with extended partial hepatectomy. The patient then received gemcitabine and capecitabine in the adjuvant setting for 3 months, which was discontinued because of toxicity. The patient underwent active surveillance for 5 years after liver resection and had a recurrence, with a surveillance scan showing left hilar lymph node enlargement and biopsy confirmed pancreatic ACC. The patient received three cycles of FOLFOX followed by two cycles of gemcitabine and nab-paclitaxel (abraxane). Afterward, the patient’s condition worsened, with progression of disease, peritoneal carcinomatosis, and ultimately death. This patient’s OS was 92 months. 

Also of note is that one of our patients was positive for SAT-B1-RET fusion. Before the fusion was identified, this patient underwent treatment with FOLFIRINOX. Since the fusion was identified, she has been receiving treatment with selpercatinib, which was approved by the U.S. Food and Drug Administration for treatment of any cancer with RET fusion/gene rearrangement [6]. At the time of writing, the patient’s disease is still responding favorably to selpercatinib, but the patient did experience the development of chylous ascites, which occurs in about 7% of patients taking selpercatinib [7]. We show the primary tumor response to selpercatinib in Figure 2.

At the time of ACC diagnosis, 12 patients had metastasis to the liver, the most common site of metastasis (N = 12/14). Two patients had distant metastasis to the lungs, pleura, and supraclavicular lymph nodes. Only one patient underwent liver-directed therapy, which was a partial hepatectomy. This was the same patient who underwent the Whipple procedure and presented with liver metastasis 2 years later. As mentioned above, the patient went on active surveillance for 5 years and eventually had progression of disease. Fifteen patients were deemed to have unresectable ACC at the time of diagnosis. Another patient, who received FOLFIRINOX in the neoadjuvant setting, later underwent surgical resection of the primary tumor. His ACC progressed during FOLFIRINOX administration and he was treated with gemcitabine and abraxane, but he died because of disease progression. 

The median OS was 24 months for the patients with metastatic disease in our study. The median OS in the gemcitabine-based treatment group was 20.5 months, and that in the FOLFOX based group was 26 months. The OS between the above two treatment groups was compared with Gehan–Breslow–Wilcoxon test. The obtained *p*-value was 0.3346 (Figure 3A).The *p*-value with the log-rank test was 0.7151. 

We compared the progression-free survival of first-line treatment with 5-FU-based treatment vs. gemcitabine-based regimens, depicted in the Kaplan–Meir curve in Figure 4. Most likely due to our small sample size, the *p*-value (0.3754) by the log-rank test was not statistically significant (Figure 3B). Majority of the patients received platinum-based chemotherapy including FOLFOX, FOLFIRINOX, and gemcitabine-cisplatin. Only two patients did not receive platinum based chemo regimen (Table 3). 

We also included additional two cases who underwent BostonGene testing, including genomic alterations, RNA expression, and tumor microenvironment (TME) analysis. The features are summarized in Figure 4. The two cases included one case with mixed ACC and adenocarcinoma, who was found to have NTRK1 fusion (case 17#, Figure 4A), and another case with pure ACC on histology (#18). Both cases showed immune desert features (Figure 4A). Cell compositions and RNA signatures showed few T cells, and T cell trafficking and B cells were observed in the tumors (Figure 4B,C). CEACAM5 RNA expression was low in both cases compared with PDAC, and CDH6 RNA expression levels were higher than PDAC (Figure 4D). CD47, MMP1, and CD74 RNA levels showed lower trends than the reference PDAC data (Figure 4D). FAP expression showed opposite trends in the two cases: with one sample lower than PDAC (18#, pure ACC) and the other one higher than PDAC references (17#, mixed ACC-adenocarcinoma, Figure 4D). Patient 17# was treated with larotrectinib for 15 months until progression of disease.

## 4. Discussion

In this case series, we summarized the clinical and molecular features of 16 ACC patients. We found that all of the patients who were tested for *KRAS* mutations were negative (15/15) and 5 of 12 (42%) of the patients who underwent DNA damage repair gene testing had germline and/or somatic mutations in DNA damage repair genes (*BRCA1/2*, *PALB2*, and *ATM*). One patient had RET fusion and has been on a RET inhibitor for more than 42 months. One of the two additional cases who underwent BostonGene testing also had NTRK1 fusion, which was treated with larotrectinib and the patient did not have progression of disease until 15 months later. Our study suggests the importance of molecular testing for ACC to identify actionable genomic alterations. 

ACC is typically seen in individuals above the age of 60 and predominantly occurs in male patients, with a male-to-female patient ratio of 3.6 [1,8]. Our population demographics were similar, and most of the patients were male (81%, Table 1). Other authors have described lipase hypersecretion syndrome, a type of paraneoplastic syndrome that makes up the Schmid triad with multifocal fat necrosis and polyarthralgia, in about 10–15% of patients with pancreatic ACC [1,9]. Most of our patients presented with abdominal pain (N = 12, 75%) or obstructive jaundice (N = 3, 19%) and ACC was an incidental finding in the remaining one patient. We did not observe paraneoplastic syndrome in our limited population. Consistent with another report, liver metastasis was the most common metastatic site in our patients (N = 10/14) [2]. The role of liver-directed therapy in ACC is uncertain. No significant survival was found with tumor debulking surgery in patients with metastatic disease [10]. However, long-term survival after aggressive surgery of primary tumor and liver metastases resection in pancreatic ACC was reported [11]. One of our patients presented a significant OS of 92 months. This patient underwent the Whipple procedure and removal of metastatic liver lesions. Even if no conclusions can be drawn from a single case, this patient’s cancer treatment history potentially points to improved progression-free survival after surgery in selected ACC patients who undergo resection of metastatic lesions. The selection of such patients who could benefit from aggressive surgery based on their clinical, biological, and molecular features remains undefined. 

Because of the rarity of ACC and its morphological resemblance to normal pancreatic acini, diagnosis can be challenging. ACC also shares histological features with neuroendocrine carcinoma, pancreatoblastoma, and mixed tumors [12]. Mixed acinar-endocrine carcinomas are histologically similar to the pure ACCs but have more than 25% endocrine cells [13]. Immunohistochemical staining is often required to confirm the diagnosis of ACC, as it helps to differentiate this tumor from other pancreatic neoplasms with similar morphological features [14]. Trypsin, chymotrypsin, lipase, amylase, and carboxyl ester lipase are some of the exocrine pancreatic enzymes used to differentiate ACC from other pancreatic tumors [15]. These enzymes are expressed at varying degrees in pancreatic ACC, so they may not be expressed in all of these tumors. Antibodies against trypsin and BCL10, which identify carboxyl ester lipase, have been most sensitive in immunohistochemical staining of ACCs [16]. ACC was reported to be strongly positive for trypsin and chymotrypsin and negative or focally positive for synaptophysin and chromogranin [13]. In our study, all patients tested were positive for trypsin (N = 9/9) and chymotrypsin (N = 2/2). Five of twelve patients were positive for synaptophysin and three of thirteen were positive for chromogranin. One patient had mixed ACC with NET. Two patients tested positive for BCL-10 (2/2) and BCL-2 (2/2). 

Identification of the genetic alterations in ACC is key to further exploration of targetable mutations for treatment. *KRAS* is mutated in the majority of pancreatic adenocarcinoma cases, and researchers recently showed that targeting *KRAS* in treatment of pancreatic cancer produced promising results [17,18]. None of our 16 patients tested positive for *KRAS* mutations (15 tested negative, and 1 was not tested for it). This finding is consistent with earlier reports of a lower *KRAS* mutation rate for ACC (13.6%) than for pancreatic adenocarcinoma (85.1%) [4,19,20,21,22]. As described by Kim and Knepper [23], *KRAS* WT tumors are well known to have higher actionable mutation and gene fusion rates than *KRAS*-mutated tumors. These two authors reported high gene fusion rates in patients with *KRAS* WT pancreatic ductal adenocarcinoma [23]. Their analysis of tumor tissue from almost 2500 patients with pancreatic cancer revealed that those lacking mutations of the *KRAS* gene (*KRAS* WT) frequently harbored mutations of genes associated with various critical cellular functions. These include DNA damage repair genes such as *ATM*, *BAP1*, *BRCA2*, *FANCE*, *PALB2*, and *RAD50*. In another study, Singh et al. [24] described the oncogenic drivers for *KRAS* WT pancreatic cancer. Of the 795 samples of pancreatic cancer, mostly PDAC, they analyzed, including samples from five cases of ACC, 9.2% were WT for *KRAS*. The mitogen-activated protein kinase pathway was a key oncogenic driver for *KRAS* WT pancreatic cancers. Also, the researchers identified *BRAF* mutations and receptor tyrosine kinase fusions to be possible targetable mutations in this population. 

Of note, the highest mutation percentage we identified in our patient population was that for *NF1* and *RB1* (28%, Figure 1). Ramakrishnan et al. showed that for pancreatic ductal adenocarcinoma patients who tested wild type for *KRAS*, inactivation of the *NF1* gene played a vital role in oncogenesis by triggering acinar-to-ductal metaplasia and pancreatic cancer development in situ [25]. The role of the *NF1* gene in pancreatic ACC needs to be explored further. Also, La Rosa et al. [26] described the role of *TP53* mutation in pancreatic cancer. Their study revealed *TP53* positivity in 13% of primary ACCs and 31% of metastases. They concluded that *TP53* positivity may be associated with tumor progression and shortened survival. In our study, only one patient presented *TP53* positivity. It should be noted that not all patients were tested for all possible gene mutations by the limited gene panels, suggesting that the number presented may be an underestimation.

In our patients, somatic and/or germline mutations of DNA damage repair genes (*BRCA1/2*, *PALB2*, and *ATM*) were present in 5 of 12 patients (42%) tested for at least some of these genes (Figure 1). High germline and somatic DNA damage repair gene mutations have been well documented in ACC with mutation rates up to 45% by whole-exome sequencing [2,4,21,22,27,28]. Pure pancreatic ACCs have a higher prevalence of germline *BRCA1*, *BRCA2*, and *PALB2* pathogenic variants (42%, N = 13/31) than ACCs with mixed with ductal or neuroendocrine histology (11%, N = 2/18), based on a recent report with mostly *BRCA2* (35%, N = 11/31) [28]. Others also reported high rates of homologous recombination-related gene mutations in ACC cases compared to pancreatic ductal adenocarcinoma (PDAC), including mutations of *BRCA2* (13.6%), *BRCA1* (2.3%), and *ATM* (11.4%); they also reported that 25% had mutations of at least one of five genes (*ATM*, *ATR*, *BRCA1*, *BRCA2*, and *PALB2*) [4]. Furthermore, these authors reported a higher response rate for FOLFIRINOX (53.8%) than for gemcitabine and nab-paclitaxel (23.5%) in ACC patients, according to a search of the Japanese Nationwide Comprehensive Genomic Profiling database [4]. A previous retrospective analysis demonstrated higher response rates for platinum-containing (40%) and irinotecan-containing (29%) regimens than for gemcitabine-containing (7%) regimens [29]. It also demonstrated a higher response rate for monotherapy with tegafur/gimeracil/oteracil (17%) than for monotherapy with gemcitabine (3%) [29]. Other researchers also observed higher response rates for treatment with platinum-based regimens in ACC patients with homologous recombination-related gene mutations in several studies [30]. Recently published data from the Sun-Yat-sen University Cancer Center by Xu et al. of a pooled analysis including 32 studies and a total of 86 patients with acinar cell carcinoma compared first-line therapy with fluoropyrimidine-based vs. gemcitabine-based treatments. The ORR of the fluoropyrimidine-based group (59.6%, 28/47) was higher than that of the gemcitabine-based group (15.4%, 6/39) (*p* < 0.001) [31]. As seen in the comparative analysis of treatment and outcome (Table 3), the improved response of our patients to 5-FOLFOX-based regimens (median OS 26 months) than to gemcitabine-based regimens (median OS 20.5 months) was not statistically significant (*p* = 0.3346). This might be due to our relatively small sample size. A larger study by Sakakida et al. of patients with rare subtypes of pancreatic cancer, including 44 patients with ACC, demonstrated a higher overall response rate for FOLFORINOX than for gemcitabine-based treatment (61.5% vs. 23.5%; *p* = 0.06) and a significantly longer median time to treatment failure for the former than for the latter (42.3 weeks vs. 21.0 weeks; *p* = 0.004) [4]. Only one patient had a response after receiving gemcitabine/nab-paclitaxel combined with cisplatin and all of the other four patients who received gemcitabine-based regimens did not respond. The results implied a preference for FOLFIRINOX/platinum chemotherapy in ACC. 

None of our patients had PARP inhibitors as part of their treatment. The role of PARP inhibitors like olaparib has been well established in pancreatic adenocarcinoma patients with germline BRCA mutations after treatment with platinum-based regimens [32,33]. But the efficacy of PARP inhibitors in acinar cell carcinoma has been primarily described in case reports only. One case report from Shanghai mentions the use of olaparib as a third-line therapy, with a brief partial response of tumor shrinkage [34]. Another case report from France describes a patient treated with olaparib who had stable disease for almost 30 months, another case report describes a patient has been on olaparib maintenance for 24 months with partial response, and another case report from Japan described a patient who also had partial response and continues to be on maintenance at 46 months [35,36,37]. These cases highlight the need for further study on utilizing PARP inhibitors for pancreatic acinar cell carcinoma patients with germline BRCA-2 mutations.

Among our patients, one had RET fusion (specifically, SAT-B1-RET fusion). This patient is undergoing treatment with selpercatinib for more than 42 months. Targetable fusions like RET fusion may open more avenues for treatment in the ACC patient population [7,38,39]. Gene rearrangements such as *BRAF* and *CRAF* fusion were reported in 23% of patients in a 44 patient case series including 16 pure ACC, 14 mixed acinar/neuroendocrine, and other histologies [27]. The study revealed the diversity of the BRAF breakpoints and fusion partners. Others also reported up to 30% fusion genes affecting *BRAF*, *CRAF*, *RET*, and *NTRK1/2/3* in ACC [28]. Interesting, the RAF gene fusions were mutually exclusive with the inactivation of DNA repair genes (45%) and the *BRAF* oncogenic alterations were exclusively found in non-DNA damage repair gene-mutated pure ACCs [27,28]. The studies suggested the importance of MAPK pathways and DNA damage repair genes. Authors have reported chromosome 11p loss to be the most frequent genomic alteration in ACC patients (50% (6/12)), with the APC/β-catenin pathway gene mutation occurring frequently (23.5% (4/17)) [40]. Also, investigators found *CTNNB1* mutations in 13.6% of ACC cases [4] and mismatch repair deficiency in 14% of ACC cases according to immunohistochemistry [41]. We did not find chromosome 11p loss nor *CTNNB1* mutation, likely due to the limited sample size. 

In the two additional cases who underwent comprehensive molecular profiling using BostonGene tests, including DNA and RNA sequencing and TME analysis, one of the patients was found to have *NTRK1* fusion (Figure 4A). NTRK1/2 and 3 gene fusions are known oncogenic drivers in solid tumors. Inhibitors targeting NTRK1 fusions, like larotrectinib, were approved by FDA for solid tumors with NTRK1 fusion, and in the phase I/II clinical trials, larotrectinib showed an overall response rate (ORR) of 75%, with 95% CI, 61 to 85 [42,43]. Our patient was given larotrectinib without progression of disease until the 15th month. It is important to mention the newer NTRK1 inhibitors, like repotrectinib and entrectinib, are also being tested out in solid tumors. More research is needed to see the efficacy of NTRK1 inhibitors in pancreatic cancer, especially acinar cell carcinoma [44]. The results described above demonstrate the importance of molecular profiling and gene fusion testing of ACC patients to create more personalized treatment plans. The National Comprehensive Cancer Network guidelines currently recommend tumor genomic profiling (e.g., NGS) to guide precision-based and targeted approaches to cancer treatment. The U.S. Food and Drug Administration has approved DNA-based assays for mutation analysis. These assays are effective in evaluating mutation landscapes but accurately identifying gene fusions and exon-skipping events, especially novel ones, remains a challenge [45]. Whereas NGS reliably detects single-nucleotide variants and small insertions/deletions, it is less reliable for detecting larger structural variants such as chromosomal rearrangements, including fusions and copy number variations. Fusion breakpoints occurring within introns or repetitive regions in DNA negatively impact assay performance because of the size limitations of hybridization-capture probes [45]. NGS often requires relatively sizeable quantities of high-quality DNA or RNA, which can be challenging to obtain, especially from archival or degraded samples for RNA-based assays. RNA-based assays are more effective in identifying gene fusions and exon skipping. This poses a unique challenge, however, as RNA is unstable, has varying expression levels, and lacks a double-stranded context [46]. The other layer of complexity includes the diversity of breakpoints of partners of gene fusions, and the limited gene panels used in clinical testing may not be able to capture the full spectrum of tumor somatic gene rearrangements without whole-exome sequencing or whole-genome sequencing. Advancement of clinical testing platforms is warranted. Being able to identify targetable fusions in molecular testing is of the utmost importance, as it provides the opportunity for individualized treatment approaches for ACC and other *KRAS* wild-type tumors based on molecular profiles. Hence, we recommend both DNA- and RNA-based fusion panel tests to increase the chance of identifying targetable genomic alterations, especially gene fusions, in ACC. This approach may lead to more effective therapies tailored to the specific genetic alterations driving ACC. 

The features of the two cases who underwent genomic analysis, RNA expression, and tumor microenvironment (TME) analysis are summarized in Figure 4. One case had mixed ACC and adenocarcinoma (NTRK1 fusion positive, #17) and the other case had pure ACC on histology (#18, Figure 4A). Both tumor tissues showed immune desert features with very few immune cells, such as T cells and B cells (Figure 4A–C). CD47, MMP1, and CD74 RNA levels showed lower trends than the reference PDAC data (Figure 4D). PDAC is well known for an immune suppressive TME with overexpression of CD74, CD47, and stromal MMP signals [47,48,49]. Effective therapeutics to overcome the immune resistance in PDAC remains challenging [50,51]. Profiling patients’ samples using comprehensive platforms, including TME analysis, in clinical practice and correlative studies under clinical trials could be critical to understanding the resistance mechanisms [52]. The *CEACAM5* gene encodes the tumor marker carcinoembryonic antigen (CEA), and CEACAM5 is commonly overexpressed in PDAC and associated with epithelial–mesenchymal transition and poor prognosis [53]. Targeting CEACAM5 is under investigation in gastrointestinal cancer, and its role in pancreatic cancer remains unclear [54]. CEACAM5 RNA expression was lower in both cases compared with PDAC (Figure 4D). It was reported previously that CEA is not a sensitive tumor marker for ACC and only 15% of cases had elevated CEA levels [55]. CDH6 RNA expression levels were higher than PDAC (Figure 4D). CDH6 is expressed in cholangiocarcinoma, gastric cancer, renal cell carcinoma, ovarian cancer, and other cancers but less studied in pancreatic cancer, including ACC [56,57,58]. FAP expression showed opposite trends in the two cases: with one sample lower than PDAC (#18, pure ACC) and the other one higher than PDAC references (#17, mixed ACC-adenocarcinoma, Figure 4D). FAP is highly expressed in cancer-associated fibroblasts (CAFs) and critical in PDAC [59]. Less is known about the roles of gene expression in pancreatic ACC. The sample sizes here are small and more studies at the RNA level and TME analysis in ACC would be helpful to understanding the molecular and immune features of ACC. 

The limitations of the present study include that it is a single-institution case series with potential population bias and limited numbers of patients. The molecular testing panels, which evolved over the years, included expanded gene panels over the years in different testing platforms and variations in gene fusion panels. Thus, different genes were tested among our patients. For example, *KRAS* mutation testing was not routinely performed in one case diagnosed in 2009, which was early in the era of mutation testing for pancreatic cancer, and not all patients were tested using the DNA- and/or RNA-based gene fusion panels. Only two patients underwent RNA sequencing and TME analysis using the BostonGene testing platform.

## 5. Conclusions

Herein, we describe the molecular and clinical course of pancreatic ACC in 16 patients. No patients had *KRAS* mutations, *HRR* gene mutation rates were high (5/12, 42%), and one patient with SAT-B1-RET fusion treated with a RET inhibitor had OS of more than 42 months. We also observed a tendency toward better OS with FOLFIRINOX than with other treatment regimens (median OS duration, 26 months versus 20.5, *p* = 0.3346). In summary, our study shows the clinical and molecular features of ACC and suggests the use of molecular profiling and gene fusion panels in the treatment of ACC.

## Figures and Tables

**Figure 1 cancers-16-03421-f001:**
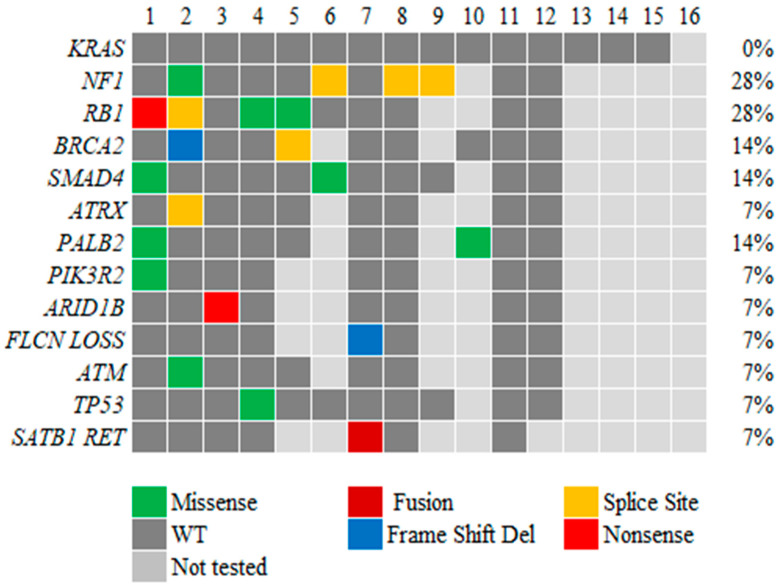
Oncoplot of genomic alterations identified in the study patients. Del, deletion.

**Figure 2 cancers-16-03421-f002:**
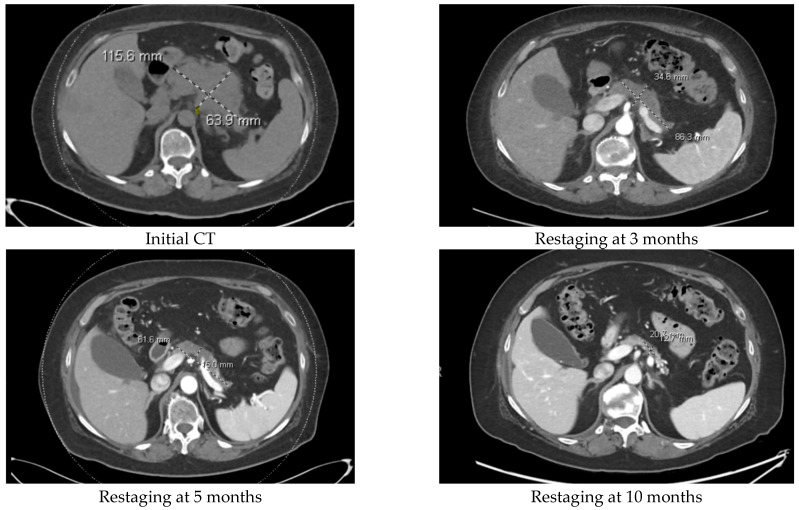
Patient with RET-1 fusion gene, primary tumor response to selpercatinib.

**Figure 3 cancers-16-03421-f003:**
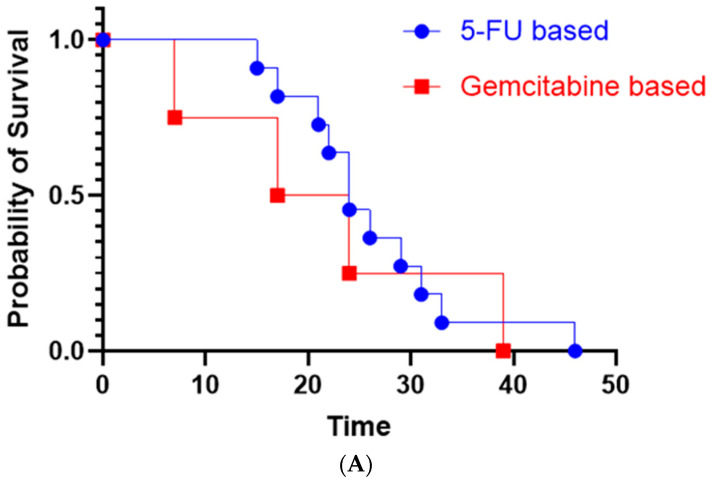
(**A**) Kaplan–Meir curves comparing overall survival of 5-FU-based vs. gemcitabine-based treatment regimens. (**B**) Kaplan–Meir curves comparing progression-free survival of first-line 5-FU-based vs. gemcitabine-based treatment regimens.

**Figure 4 cancers-16-03421-f004:**
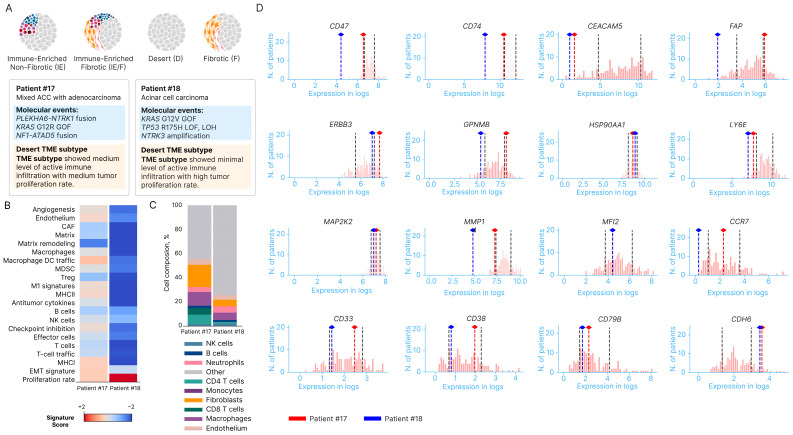
RNA expression and TME features for the two cases tested by BostonGene. (#17, mixed ACC with adenocarcinoma, found NTRK1 fusion) and #18 (pure ACC). (**A**) TME features of both cases revealed immune desert type. (**B**,**C**) RNA expression signatures and % of cell composition. (**D**) Relative expression of selected genes (dashed lines for reference ranges of PDAC).

**Table 1 cancers-16-03421-t001:** Demographic characteristics of the 16 study patients.

Characteristics	Number of Patients (%)
Sex	
Male	13 (81)
Female	3 (19)
Median age at diagnosis, years	62.5
Race	
White	14 (88)
Hispanic	1 (6)
Asian	1 (6)
Primary tumor location	
Head of pancreas	8 (50)
Body of pancreas	5 (31)
Tail of pancreas	2 (13)
Body and head of pancreas	1 (6)
Tumor status at initial diagnosis	
Resectable/borderline resectable	2 (12)
Metastatic	14 (88)
Smoking status (current/former smoker)	9 (56)
Alcohol status (current/former drinker)	10 (63)

**Table 2 cancers-16-03421-t002:** Histology and immunohistochemistry results for the 16 study patients.

Characteristic	Number of Patients
Histology	
Pure acinar cell carcinoma	14
Mixed acinar-neuroendocrine	1
Mixed acinar and adenocarcinoma	1
Marker positivity	(N of positive/tested)
Trypsin	9/9
Chymotrypsin	2/2
Synaptophysin	5/12
BCL-10	2/2
Cytokeratin 7	8/9
Alpha 1-antichymotrypsin	2/2
Chromogranin	3/13
Alpha 1 antitrypsin	1/2
BCL-2	2/2

**Table 3 cancers-16-03421-t003:** Treatment Response: Abbreviations: mo—month; rx—treatment; yrs—years; Gem—Gemcitabine; Neoadj—Neoadjuvant; Adj—Adjuvant; Xelox—Capecitabine/Oxaliplatin; Folfox—5-fluorouracil, leucovorin, and oxaliplatin; Folfiri—5-fluorouracil, leucovorin, and irinotecan; Folfirinox—5-fluorouracil, leucovorin, oxaliplatin, and irinotecan; Avastin—Bevacizumab; Onivyde—nano liposomal irinotecan; SD—stable disease; PR—partial response; POD—progression of disease; DOD—dead of disease; b/l—bilateral; UE—upper extremity; HFM—hand foot mouth disease; LTFU—lost to follow-up; OS—overall survival in months.

	First Line	Second Line	Third Line and Beyond	OS
67/M	Gemcitabine/nab-paclitaxel/Xeloda POD at 5 mo.	Pt DOD prior to next line rx.		7
70/M	FOLFOX 3 mo followed by Xelox 1 mo and XRT to liver and then surveillance until POD at 18 mo.	Xelox, after 2 cycles switched to FOLFOX due to severe HFM for 1 mo with POD.	Everolimus, SD until POD in 10 mo. Gem/Abraxane for 2 mo with POD. 5-FU/Onivyde 3 mo with POD and DOD.	46
72/M	Folfirinox 4 mo with PR and lost to follow-up.			15
61/M	Folfirinox 3 mo and Folfiri 3 mo, maintenance Xeloda for 6 mo with PR 3 mo, 6 mo, and 9 mo, and POD at 12 mo.	Gem/Abraxane 2 mo with POD.	Folfiri 2 mo with POD, clinical trial recommended, and LTFU.	24
59/M	Xelox for 6 mo with partial response at 3 and 6 mo, maintenance Xeloda for 2 mo with PR and LTFU.			22
53/M	Folfox 2 mo with POD.	Folfirinox for 2 mo with POD.	Gem/Abraxane for 3 mo with SD. Xeloda added, PR at 2 mo and 4 mo, POD at 6 mo and DOD.	29
58/F	Folfirinox 1 mo switch when RET-1 gene fusion identified.	Selpercatinib. Continuing to show PR.		26
52/M	Folfox 2 mo with POD.	Xeloda + XRT with PR f/u by surveillance for 3 mo, but POD with new mets.	FOLFIRINOX 2 mo with POD. Gem/Abraxane with SD at 2 mo and 6 mo, POD at 8 mo. Erlotinib + Avastin 3 mo with POD. Pembrolizumab 1 mo with POD and DOD.	24
65/F	Gemcitabine/cisplatin 4 mo with POD.	Folfox 2 mo with POD.	Gem/Abraxane 4 mo with PR, and added, Xeloda 9 mo with PR at 3 mo, and SD at 6 and 9 mo. LTFU.	39
49 /M	Folfirinox for 7 mo with PR at 3 mo, SD at 5 and 7 mo, Xeloda maintenance for 24 mo with PR.			31
70/F	Surgery, no neoadj or adj chemo, on surveillance for 3 yrs.	Recurrence—surgery adj chemo Xeloda 3 mo and surveillance for 5 yrs.	Recurrence as peritoneal carcinomatosis rxed with Gem/Abraxane with rapid POD and pt DOD.	92
66/M	Neoadj Folfirinox 4 mo followed by Xeloda + XRT and surgery followed by adj chemo folfirinox 2 mo.	Recurrence with pleural mets rxed with Gem/Abr 2 mo with POD and phase I referral. DOD		21
59/M	Xelox 1 mo, b/l UE thrombosis, and lost central IV access.	Gem/Xeloda 1 mo + Xeloda XRT, POD at 2 mo.	FOLFIRI 4 mo with POD and reference to phase I. DOD	17
62/M	Gemcitabine/Cisplatin 3 mo with POD.	Gemcitabine/Erlotinib with POD.	Phase 1, SD for 11 mo followed by POD and DOD.	17
66/M	Folfirinox 5 mo with SD at 3 mo, POD at 5 mo.	Gem/Tazarotene/xeloda for 5 mo, SD—Rx break 6 mo per pt preference, restaging with POD. LTFU.		33
55/M	Neoadj Gemcitabine + Abraxane + Cisplatin 2 mo f/u by surgery—prolonged post-op recovery and post-op restaging with early relapse at 5 mo.	Adj FOLFIRINOX 3 cycles (1 mo) with POD.	Gem/Abraxane/5-FU and Cisplatin for 5 mo with SD at 2 mo and PR at 5 mo. Erlotinib added POD 7th mo and referral to Phase I. POD within 3 mo in Phase I. DOD.	24

## Data Availability

Individual patient level data are not publicly available to maintain compliance with HIPPA regulations and IRB protocol. Anonymized data are available for non-commercial use from the corresponding author upon request, pending data usage agreement and/or IRB-approved collaboration.

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
