# Peer review of "Molecular and Clinical Features of Pancreatic Acinar Cell Carcinoma: A Single-Institution Case Series"

_cancers, 2024, doi:10.3390/cancers16193421_

Round 1
Reviewer 1 Report
Comments and Suggestions for Authors
Dr. Pillai et al present a single center case series on pancreatic acinar cell carcinoma (ACC) patients, detailing their molecular and clinical features. Data on ACC are scarce due to its rarity, particularly with respect to molecular profiling. My concerns are as follows.
1. Details on the molecular testing performed should be provided, including the specimen used for testing. Liquid panels will have lower sensitivity for KRAS mutations.
2. Table 1 should include key laboratory figures including tumor markers; ACC is associated with lower CA19-9 levels and high AFP and lipase levels. Table 1 could also be visually improved by, for example, left justifying the text and using indents to clarify categories.
3. Did patient #16 in Figure 1 undergo molecular testing? Figure 1 shows that none of the genomic alterations were tested.
4. Table 3: Please revise the table so that it is easier to read and understand. The lines overlap and it is difficult to tell whether information relates to patient 15 or patient 16, for example. Please also write out fuller phrases. What do “stability”, “stable response”, and mixed response” mean? It would be easier to understand if best response and PFS were provided for each line. Also, consistently use “months” instead of “cycles”.
5. Table 3: Patient 1: “Xelox 3 cycles with stable disease, rx switch due to loss of vascular access” is confusing. If the authors mean CV access, there was no use of 5-FU in this patient. If peripheral IV insertion was an issue, then it is unclear how gemcitabine was used as second-line treatment.
6. Table 3: 6 patients went on to clinical trials. Please confirm that the authors are permitted to disclose OS information on these patients.
7. In the discussion, the authors state: “As seen in the comparative analysis of treatment and outcome (Table 3), the improved response of our patients to 5-FOLFOX–based regimens (median OS 26 months) than to gemcitabine-based regimens (median OS 20.5 months) was not statistically significant (p=0.3346)”. Several points: 1) I would prepare platinum-based vs. non-platinum based regiments, not FOLFOX-based vs GEM-based because GEM-based regimens may also contain platinum agents. 2) it may be preferable to compare first-line PFS, for example, given that several patients went on to clinical trials with no disclosed information on what agents were given. 3) Kaplan-Meier curves may be presented as a Figure.
8. Five patients had DNA damage repair gene mutations. Were PARP inhibitors used in any of the cases? A quick discussion on the potential role of PARP inhibitors in ACC may be warranted in the discussion section, although it is already very long.
9. Please include explanations of any associations between histology/immunochemistry findings and genetic alterations or outcomes with platinum-based regimens that were found in in this study, if any.
Reviewer 2 Report
Comments and Suggestions for Authors
The paper titled "Molecular and clinical features of pancreatic acinar cell carcinoma: a single institution case series" by Balachandran Pillai provides a retrospective analysis of 16 patients with pancreatic acinar cell carcinoma (ACC) treated at MD Anderson Cancer Center, focusing on clinical and molecular characteristics, treatment outcomes, and the implications for personalized treatment strategies.
The study's strengths lie in its identification of key genetic alterations, including actionable mutations like RET and NTRK1 fusions, and a high prevalence of DNA damage repair gene mutations (e.g., BRCA1/2, PALB2, ATM). The discovery of these mutations in ACC patients contributes to the understanding of the genetic basis of this rare cancer and highlights the potential for targeted therapies. For instance, one patient with a RET fusion treated with selpercatinib demonstrated a favorable response for over 42 months, showing the promise of personalized medicine. The study also notes trends in better overall survival with FOLFIRINOX compared to gemcitabine-based regimens and includes novel insights into the tumor microenvironment, such as the immune desert characteristics in ACC.
However, the study has some limitations.
Major points:
Recent literature already provides valuable insights into ACC, including the absence of KRAS mutations and the efficacy of FOLFIRINOX, which is more effective than gemcitabine-based treatments for metastatic PACC, and the higher prevalence of homologous recombination-related genes, including BRCA1/2, diminishing the novelty of this finding. This is discussed in reference 4 in the manuscript. The important reference by Xu et al. (PMID: 35509769) has not been mentioned. This study, combining data from 22 ACC patients from Sun Yat-sen University Cancer Center and a pooled literature analysis, found that fluorouracil-based regimens, particularly FOLFIRINOX, are more effective than gemcitabine-based treatments for metastatic PACC. Additionally, maintenance therapy with PARP inhibitors may benefit patients with BRCA mutations following effective platinum-containing chemotherapy (PMID: 35509769).
While the study identifies various genetic alterations, it does not explore their direct impact on therapeutic outcomes beyond the RET fusion case. DNA damage repair gene mutations are also not discussed with respect to potential treatment outcomes, particularly regarding PARP inhibitors and platinum-based therapies.
Additionally, the interesting case involving the NTRK1 fusion is not pursued, missing an opportunity to discuss the potential benefits of NTRK inhibitors like larotrectinib or entrectinib, which are relevant in today’s precision oncology landscape.
Table 3 is disorganized and lacks clarity regarding the text for the third and beyond therapy columns. It is necessary to improve this table and to link the final genetic mutations to it. The content of Figure 1, which shows genomic alterations, does not match the content in Table 3. For instance, in Figure 1, patient #7 is shown as the one with the RET fusion. In the text (line 165), it is stated that the patient, who has been receiving treatment with selpercatinib, is female. However, in Table 3, patient #7 is described as a 55-year-old male.
Minor points:
Please also show the p-values for the log-rank and Tarone tests in addition to the Gehan-Breslow test (for early survival differences) in the comparison of survival outcomes for FOLFIRINOX vs. gemcitabine.
In conclusion, apart from the case with the RET fusion, the manuscript does not provide new insights into the molecular and clinical features of ACC, particularly as it does not sufficiently discuss the potential for actionable mutations to guide targeted therapies.
However, the one case demonstrating the RET fusion and the administered targeted therapy is still of interest for personalized medicine and should be published as a case report.
The study requires a more detailed exploration of the impact of these mutations on therapeutic outcomes and a graphical illustration of potential precision oncology decisions for future patients. Specifically, the unique RET fusion case should be illustrated in an additional figure, with a time course of the treatments and clinical images showing the favorable response, to make it more appealing for clinicians.
Therefore, a major revision is recommended to address these issues and enhance the paper's contribution to the field.
Round 2
Reviewer 1 Report
Comments and Suggestions for Authors
The authors have adequately revised their manuscript. I have no further comments.
Reviewer 2 Report
Comments and Suggestions for Authors
The manuscript has improved.